



# Functional Specifications and Testing Requirements of Grid-Forming Offshore Wind Power Plants

Sulav Ghimire[1,2], Gabriel M.G. Guerreuro[1,2], Kanakesh Vatta Kkuni[1], Emerson D. Guest[1], Kim H. Jensen[1], Guangya Yang[2], and Xiongfei Wang[3]

[1]Siemens Gamesa Renewable Energy A/S, 7330 Brande, Denmark
[2]Technical University of Denmark, 2800 Kongens Lyngby, Denmark
[3]KTH Royal Institute of Technology, Stockholm, Sweden

**Correspondence:** Sulav Ghimire (sulav.ghimire@siemensgamesa.com)

**Abstract.** Throughout the past few years, various transmission system operators (TSOs) and research institutes have defined several functional specifications for grid-forming (GFM) converters via grid codes, white papers, and technical documents. These institutes and organisations also proposed testing requirements for general inverter-based resources (IBRs) and specific GFM converters. This paper initially reviews functional specifications and testing requirements from several sources to cre-
ate an understanding of GFM capabilities in general. Furthermore, it proposes an outlook of the defined GFM capabilities, functional specifications, and testing requirements for offshore wind power plant (OF WPP) applications from an original equipment manufacturer (OEM) perspective. Finally, this paper briefly establishes the relevance of new testing methodologies for equipment-level certification and model validation, focusing on GFM functional specifications.

## 1 Introduction

Ongoing rapid growth of inverter-based resources (IBRs) in modern power systems leads to a significant loss of system inertia and short-circuit power, which is followed by several challenges such as voltage, frequency, and synchronisation instability (Milano et al., 2018). The need for newer technologies, such as grid forming (GFM) technology, has become inevitable in maintaining and improving power system stability. The recent rapid growth in wind generation, including offshore wind power (IEA, 2023, 2022; Barthelmie and Pryor, 2021), also fosters the rise in large-scale offshore wind power plants (OF WPPs). As

part of the major power source, GFM converter control technology must be integrated into the WPPs to enhance power system stability.

Existing OF WPPs (or IBRs in general) are dominated by grid-following (GFL) converter control technology, which synchronises itself to the grid via phase-locked loops (PLLs) and follows the frequency and voltage reference of the grid while injecting a constant power via a controlled current. Consequently, GFL converters have several issues in their synchronisation

stability, transient responses, weak grid operation, and grid support functionalities, to name a few (Aljarrah et al., 2024). On the other hand, GFM converters can define internal voltage and frequency and can behave as a voltage source as opposed to the current source behaviour of the GFL converters (Matevosyan et al., 2019) and enhance the overall stability of an interconnected system (Pattabiraman et al., 2018) as well as an islanded system (Verbe et al., 2021). In light of this, the grid-forming feature





is considered an attractive solution for OF WPP. However, there is a lack of clarity on what functionalities and performance
requirements are expected of a GFM implementation in OF WPPs.

Studies are necessary to understand the GFM control method's applicability to power grids further. The most reliable form
of such studies is via field tests. To have ample field tests of GFM control, GFM converters must be integrated into the grid,
which requires grid-code requirements defined by transmission system operators (TSOs). TSOs prepare the grid codes based on
their experience while considering suggestions from different stakeholders such as original equipment manufacturers (OEMs),
equipment vendors, and developers. To provide ample relevant recommendations to the TSOs to draft such grid codes, OEMs,
vendors, and developers require further field tests, thus making this entire process a circular "chicken and egg" problem as
termed by an ESIG (Energy Systems Integration Group) task force (ESIG, 2022). Our approach can break the "chicken and
egg" cycle by proposing performance specifications for GFM OF WPPs which are, based on the knowledge of the current
state-of-the-art, mandatory performances for GFM behaviour. Further, we suggest a set of optional performance specifications,
which existing GFL or GFM OF WPPs could provide. We also propose a set of advanced performance specifications for GFM
OF WPPs, which require significant hardware changes, technological development, and experience in the field. In order to
test the performance specifications, we also propose rudimentary testing guidelines and provide an overview of emerging test
setups.

The paper layout is as follows: Section 2 summarizes the functional specifications of GFM converters provided by various
TSOs and research institutes. Section 3 points out the capabilities and limitations of OF-WPPs and adapts and reclassifies the
GFM performance specifications for OF-WPP applications. Section 4.1 summarizes the GFM performance specifications and
recommends tests to assess them, and Section 4.2 provides an outlook of different next-generation test-benches which could
be utilized for GFM functionality testing.

The key contributions of this paper are summarised below:

– Review of GFM functional specifications provided by thirteen different sources.

– Adaptation of different GFM characteristics and functional specifications for weakly connected OF WPPs and their
  classification into mandatory, optional, and advanced requirements.

– Formulation of different test recommendations for individual mandatory, optional, and advanced GFM functional speci-
  fications.

## 2 Grid Forming Functional Specifications

All electric power generators connected to the power grids must comply with a set of performance requirements known as grid
codes and should exhibit specific performances for different testing requirements for various scenarios. For novel IBRs such as
WPPs, battery energy storage systems (BESS), and solar PV generations, to name a few, specialised grid codes and performance
requirements are needed as general requirements are not adequate for such generation sources. Furthermore, different control
methods could be applied for IBRs, such as GFL and GFM control methods, which exhibit different natures and dynamics





during operation, thus reinforcing the need for specialised performance and testing requirements. This section reviews existing technical documents, white papers, and grid codes for GFM converters, discusses and critiques the provided specifications and requirements, and summarises them. For the ease of readability, a summary of the proposed GFM requirements from these different technical documents, grid codes, and white papers are provided in Figure 1.

| Voltage source behavior/slow changing internal voltage phasor | Inherent voltage jump reactive power response | Fast fault current contribution | Inertia contribution/ RoCoF requirement | Islanding and auto re-synchronization |
|---|---|---|---|---|
| • ENTSO-E<br>• GB-GF-GC0137<br>• OSMOSE<br>• UNIFI<br>• 4-TSO<br>• NERC<br>• AEMO<br>• FinGrid<br>• InterOPERA | • GB-GF-GC0137<br>• OSMOSE<br>• UNIFI<br>• EG-ACPPM<br>• IEEE Std 2800<br>• NERC<br>• AEMO<br>• FinGrid<br>• InterOPERA | • ENTSO-E<br>• GB-GF-GC0137<br>• OSMOSE<br>• 4-TSO<br>• EG-ACPPM<br>• NERC<br>• AEMO<br>• FinGrid<br>• InterOPERA | • ENTSO-E<br>• GB-GF-GC0137<br>• OSMOSE<br>• 4-TSO<br>• EG-ACPPM<br>• IEEE Std 2800<br>• NERC<br>• AEMO<br>• InterOPERA | • VDE-FNN<br>• OSMOSE<br>• UNIFI<br>• EG-ACPPM<br>• NERC<br>• FinGrid<br>• InterOPERA<br>• TenneT |

| Phase jump active power/synchronizing active power | Sink for harmonics | System black-start capabilities | Sink for imbalances | No control interactions/ interoperability |
|---|---|---|---|---|
| • GB-GF-GC0137<br>• OSMOSE<br>• UNIFI<br>• EG-ACPPM<br>• AEMO<br>• FinGrid<br>• InterOPERA | • ENTSO-E<br>• UNIFI<br>• 4-TSO<br>• IEEE Std 2800<br>• NERC<br>• AEMO<br>• InterOPERA | • ENTSO-E<br>• UNIFI<br>• 4-TSO<br>• EG-ACPPM<br>• NERC<br>• AEMO<br>• InterOPERA | • ENTSO-E<br>• UNIFI<br>• 4-TSO<br>• IEEE Std 2800<br>• AEMO<br>• InterOPERA | • ENTSO-E<br>• 4-TSO<br>• EG-ACPPM<br>• IEEE-2800<br>• NERC<br>• InterOPERA |

| Damping active power | Withstand grid SCR changes | Active power sharing/ power dispatch | Surviving the loss of last synchronous generator | Extended Inertia via Energy Reserve |
|---|---|---|---|---|
| • GB-GF-GC0137<br>• UNIFI<br>• NERC<br>• FinGrid<br>• InterOPERA | • VDE-FNN<br>• UNIFI | • UNIFI<br>• FinGrid | • AEMO<br>• FinGrid | • 4-TSO<br>• InterOPERA |

**Figure 1.** Summary of GFM converter functional specifications as provided by different TSOs and research institutes.

## 2.1 ENTSO-E's Technical Report (ENTSO-E, et. al., 2017)

European Network of Transmission System Operators for Electricity (ENTSO-E) published a technical report on the High Penetration of Power Electronic Interfaced Power Sources and the Potential Contribution of Grid Forming Converters (HPoPEIPS) in 2017 (ENTSO-E, et. al., 2017). This technical report points out the stability issues/challenges for modern power systems,





such as reduced system inertia, short-circuit levels, system split, and other conventional issues such as rotor angle stability and voltage stability. Referring to GFM converters as a potential solution to tackle such problems, a set of performance specifications are proposed for GFM converters to enhance the system stability, which includes system voltage creation, fault current contribution, sink for harmonics and imbalances, inertia contribution, and prevention of control interactions. However, no constraints on the timescale have been placed on these performance specifications, which can result in ambiguity in the GFM implementation for WPP. In addition, there is also a lack of clarity on the physical limitations that can impact the GFM nature of the WPP. For example, according to the report, since a GFM converter should, irrespective of its control technology, behave as a Thevenin-equivalent voltage source behind an impedance, GFMs need current limiting functionalities to protect the converter switches from over-current. This leads to crucial questions opening up an entire area of discussion and research: how should the current limiting be implemented without affecting the converter voltage to change rapidly, and how should the current be prioritised during such events? Furthermore, a GFM converter is required to be a sink to harmonics for frequencies below 2 kHz. However, the technical report also identifies the underlying challenge behind this requirement: an available headroom is required to inject harmonic current or provide harmonic damping during steady-state.

## 2.2 NG-ESO's Best Practice Guide and GB-GF Specifications (NG-ESO, 2021, 2023)

National Grid Electricity System Operator (NG-ESO)'s work-group consultation specifies GFM requirements such as internal voltage source behaviour, phase jump active power, voltage jump reactive power, damping active power, fast fault current contribution, inertial contribution, and RoCoF requirement (NG-ESO, 2021). Blackstart is not listed as a requirement for GFM converters; however, generation sources providing blackstart services must have GFM functionality during operation. A supporting document – Great Britain Grid Forming (GB-GF) (NG-ESO, 2023) – provides some updates on the operational design limits, such as phase jump limits, active power transients, and minimum reaction time to phase jumps with no mandates for converter oversizing. Further, (NG-ESO, 2021) also prescribe similar tests, including RoCoF, phase jumps, fault ride-through (FRT) and fast fault current injection, 3-ph faults followed by islanding, and converter power dynamics in response to grid frequency modulation based on network frequency perturbation (NFP) plots. Voltage source behaviour is considered a central element of GFM control, which is aided by the need for a physical reactor and no virtual impedance/admittance in GFM in the preceding document (NG-ESO, 2021). This requirement was later removed from the succeeding document (NG-ESO, 2023), thus facilitating the current correction/limiting via virtual admittance in the internal voltage loop of the GFM converter. With some consideration of the initial operating conditions, energy availability, and mechanical design, these GFM performance specifications could be imposed for OF GFM WPPs.

## 2.3 VDE-FNN GFM Guidelines (VDE FNN, 2020)

The German technical regulator VDE-FNN prepared guidelines which provide general requirements for stable system operation of a power system; however, it does not specify the technical requirements (VDE FNN, 2020). The document defines a list of testing scenarios to validate the GFM performance specifications. For example, tests on grid voltage phase and magnitude step, grid RoCoF response, harmonics and sub-harmonics, the negative-sequence current in the grid and its effect, islanding, and



grid short-circuit ratio (SCR) change, to name a few. These test cases are similar to the requirements proposed by the previous two literature, (ENTSO-E, et. al., 2017), and (NG-ESO, 2021). CIGRE also proposes test frameworks for HVDC and FACTS-based GFM, which includes a collaboratory approach between TSO and OEMs to GFM testing. The general recommendations of GFM requirements, test scenarios, and test frameworks are for HVDC systems and DC-connected power park modules (PPMs), which can also be adapted for OF WPPs with AC network connections. Further details on the testing specifications provided by VDE-FNN are presented in Section 4.

### 2.4 OSMOSE BESS Converter Requirements (OSMOSE, 2021)

OSMOSE – a European project led by the French TSO RTE and participated by 6 European TSOs and 33 partners – presents different GFM requirements for power converters used in BESS (OSMOSE, 2021). These requirements comprise a voltage source behaviour with a slow-changing internal voltage phasor, power-based synchronization, RoCoF withstand, under-voltage ride through (UVRT), fast current injection, and islanding capability, while stressing that energy headroom availability is crucial for GFM operation. Thus, OSMOSE requirements, in general, align with the National Grid ESO (NG-ESO, 2021, 2023) requirements with a primary focus on applications to BESS.

OSMOSE defines four types of GFM units with different capabilities apart from the core capabilities discussed earlier. The enlisted GFM unit types indicate that converter performance governs the GFM behaviour, not its control design.

(a) **Type-I GFM Unit:** Capable of standalone operation, provide system strength(V-Q support), fault current ($I_n \leq I_{fault} < 2I_n$).

(b) **Type-II GFM Unit:** Type-I GFM Unit with synchronizing power.

(c) **Type-III GFM Unit:** Type-II GFM Unit with inertial response

(d) **Type-IV GFM Unit:** Type-III GFM Unit with fault current ($I_{fault} > 2I_n$)

### 2.5 UNIFI Consortium GFM IBR Specifications (B. Kroposki et. al, 2022)

UNIFI consortium provides GFM requirements for IBRs inclusive of various generation sources, namely, solar PV, BESS, WPPs, STATCOMS, fuel cells, and BESS, to name a few (B. Kroposki et. al, 2022). The specifications are classified as follows:

(a) **Universal:**

(i) **Normal operation:** Autonomous grid support based on local measurement, dispatchability, positive damping to voltage and frequency oscillations, power sharing among generators, weak-grid operation and stability enhancement, maintaining voltage balance.

(ii) **Abnormal operation:** FRT, voltage source behaviour during asymmetrical faults, frequency response, inherent power responses to voltage magnitude and phase shifts, and islanding and re-synchronisation.





(b) **Additional:** Black-start, reduction in voltage harmonics, cyber-secure communication, secondary control of voltage and frequency.

Here, it is unclear if the inclusion of "cyber-secure communication" in the additional requirement implies the need for communication for grid-connected operation, synchronisation, and power sharing. It is a general understanding in the technical field that GFM converters need to be able to operate, fulfilling the grid code requirements without the need to communicate with each other. However, secure communication between the power plant controller (PPC) and the WTGs is necessary, irrespective of the control method.

### 2.6 Requirements by 50 Hertz, Ampiron, TenneT, and Transnet-BW (4-TSOs) (50Hertz et al., 2022)

Four German TSOs, namely 50 Hertz, Ampiron, TenneT, and Transnet-BW collaborated to summarise mandatory, additional, and optional features of GFM converters (50Hertz et al., 2022). Highly influenced by the HPoPEIPS (ENTSO-E, et. al., 2017), this report classifies GFM capabilities as:

(a) **Mandatory:** Voltage source behaviour, fast fault contribution, inertia contribution, control interaction prevention, converter stability.

(b) **Additional:** Sink for harmonics, sink for phase imbalance, additional inertia by an extended energy reserve.

(c) **Optional:** Black-start capability.

The collaboration between four different European TSOs in this paper sends a strong message that GFM IBRs are necessary for system stability. It also shows that TSOs are prepared to collaborate, contribute, and aid in the technological and policy-level development of GFM IBRs.

### 2.7 Expert Group on AC PPMs (EG-ACPPM) (EG-ACPPM, 2023)

In a report presented to the Grid Connection European Stakeholder Committee (GC ESC), the expert group on AC PPMs (EG-ACPPM) defined basic characteristics for GFM PPMs, which include the creation of system voltage, contribution to fault level, inertial contribution, and control robustness and stability (EG-ACPPM, 2023). The expert group report notes that the PPMs should be able to operate according to different network requirements such as active and reactive power provision, withstanding a blackout of 24 hours, black-start capabilities (optional, but black-start capable PPMs must be grid-forming, which is adapted from (NG-ESO, 2021, 2023)), islanding capabilities and re-synchronisation, to name a few. However, the report adds that the impact on PPMs in terms of technological and hardware changes to provide the GFM services must be evaluated. The EG-ACPPM also notes that UVRT and RoCoF withstand is essential for GFM PPMs, and studies on OVRT, phase jump active power, and voltage jump power are also necessary. In addition to the GFM requirements, the EG-ACPPM summarises generation source-specific capabilities and limitations to provide GFM response for PPMs and also includes a list of needed measures to utilise the capabilities. The capabilities and limitations presented in the report are reviewed in Section 3.1 for OF WPP applications.





### 2.8  ESIG Task Force (ESIG, 2022)

ESIG task force (ESIG, 2022) defines the GFM operational requirement and the lack of field knowledge and technological
know-how as a "chicken-and-egg" problem as one requires the other. The task force proposes an iterative process to break
the cycle by defining a target system to perform tests to determine system needs and functionalities and to implement them
after rigorous field testing, quality checks, and monitoring. Feedback from these steps updates the definition of the target
system. ESIG defines the system needs as synchronisation, voltage and frequency regulation, damping, protection, restoration,
capacity, and energy availability. It further proposes the operational requirements of GFM converters based on these system
needs. A summary of tests applicable to both GFM and GFL converters is also presented in the document, which can help define
different testing requirements for such converters for OF WPP applications. Although this report doesn't specifically mention
or summarise GFM performance specifications itself, it provides a practical roadmap that could be followed for defining such
specifications.

### 2.9  IEEE Standard 2800-2022 (2800, 2022)

IEEE standard 2800-2022 (2800, 2022) defines requirements for inverter-based generations in general without specifying the
converter control class, namely GFL or GFM. Some of these requirements, such as reactive power and voltage responses,
active power and frequency responses, power quality, and several test requirements, could be adopted readily for GFMs used
in OF-WPPs. The standard points out an important point on the interoperability of different power system components, which
will be crucial for the operation of the IBR-dominant power grids. The standard further points out that a crucial objective for
GFM IBR should be to utilise the full capabilities of IBRs in the face of evolving power systems rather than to attempt to
merely reproduce the behaviours of synchronous machines.

### 2.10  NERC GFM Operational Requirements (NERC, 2023, 2021)

The white paper by North American Electric Reliability Corporation (NERC) provides several operation requirements, in-
cluding constant or nearly constant internal voltage phasor during the transient and sub-transient time frame, operation in low
system strength, grid frequency and voltage stabilisation, re-synchronisation, FRT, and fault current contribution, and optional
black-start capabilities (NERC, 2023). However, since the US grids have a dominant solar PV generation (18% of total utility-
scale renewable energy generation in 2023 was from solar PV plants (US-EIA , 2023, Retrieved: March 18, 2024.)), NERC's
GFM converter requirements have a significant reflection of the capabilities of a DC buffer/storage devices seen in solar PV
plants. A previously published document from NERC (NERC, 2021) also summarises the GFM capabilities for general oper-
ation. This includes the operation of GFM in low system strength, frequency and voltage stabilisation, small signal stability
and power system oscillation damping, re-synchronisation (following unintentional islanding), FRT and fault-current contri-
bution within the hardware limits, and black-start capabilities for system restoration. This document addresses the numerous
challenges of GFM converters, namely the technological and resource capabilities, grid resilience, available energy headroom,
and the possibility of encountering newer stability challenges following the massive integration of GFM converters in the fu-





ture. The report also provides an overview of different GFM converter control methods and recommends that GFM control performance assessment be based on their performance, not control strategy, which is also pointed out by EPRI (EPRI, 2023).

### 2.11 Voluntary GFM Specifications by AEMO (AEMO, 2023)

Australian Energy Market Operator (AEMO) developed voluntary specifications for GFM converters and classified them into core and additional capabilities as follows (AEMO, 2023).

(a) **Core capabilities:** Voltage source behaviour, response to voltage magnitude and phase changes, improved frequency-domain dynamics with low impedance around system fundamental frequency, inertial response (with a suitable energy buffer via storage and power headroom via plant oversizing), surviving the loss of last synchronous generator.

   (b) **Additional capabilities:** overcurrent capability, black-start, power quality improvement.

    The voltage source behaviour of GFM converters, as defined by AEMO, requires GFM to have a constant internal voltage
phasor which is constant in a short time frame, i.e. during the sub-transient phases following a grid event, the internal voltage phasor of the GFM converter should change slowly or stay constant. The initial response of the GFM converters following a disturbance should start within a few milliseconds. This requirement is also consistent with the recommendations of GB-GF (NG-ESO, 2021) and FinGrid (FinGrid, 2023); the former was discussed earlier in his paper, and the latter is discussed in the following section.

### 205 2.12 Grid Forming for BESS by FinGrid (FinGrid, 2019/2020, 2023)

The TSO of Finland, FinGrid, developed grid codes for energy storage systems, typically BESS (FinGrid, 2019/2020). Following the grid-code specifications, specific study requirements were formulated for storage systems larger than 30 MW connected at 110 kV or higher (classified as type-D BESS) (FinGrid, 2023). FinGrid provides functional requirements, active power and frequency control requirements, voltage and reactive power control requirements, modelling requirements, and test require-
ments, a classification distinct from the other grid codes and performance specifications reviewed in this paper. The GFM converters are not allowed to limit the current below its capacity artificially, and they must provide GFM responses up to their rated capacity with no obligation to do so above or beyond their physical capabilities and available energy. GFM capabilities are required of the type-D BESS within the entire range of its state of charge (SoC), and grid support must be maintained even during current-limited operation. Further, type-D GFM BESS are not allowed to operate in GFL mode when connected to the
network, should resist fast changes in the internal voltage phasor in response to phase jumps, and provide near-instantaneous active and reactive power responses for frequency and voltage support in a sub-transient time frame with an initial response within a few milliseconds and a full response within 10 ms, suggesting the need for significant damping. Other GFM requirements involve a seamless transition between islanded and grid-connected modes, operation in constant active power mode and frequency-power droop mode with GFM control, and operation in constant reactive power mode and voltage-reactive power
droop mode with GFM control. Various simulation tests FinGrid defines for GFM converters will be discussed in section 4.



## 2.13 InterOPERA

A European Union-funded project on interoperability of multi-terminal multi-vendor HVDC systems was initiated in 2023, which developed functional specifications for grid-forming in HVDC converter stations and DC-connected PPMs in 2024 (InterOPERA, 2024). This functional requirement document mentions core/mandatory and optional features/functions of OF-

WPP-based GFM converters integrated into HVDC terminals. This document prepares a roadmap for a 'demonstrator' which studies the interoperability of multi-vendor, multi-terminal offshore HVDC interconnection systems with PPMs consisting of GFM or GFL converter control and explores vast areas such as interoperability, DC-FRT, and different use-cases. This document considers the inherent reactive power capability and fast fault current capability of GFM converters to be the same requirement/capability. Further, it is highlighted in the document that only capabilities unique to grid-forming need to be

classified as core capabilities. Thus, three categories of GFM performances are defined:

(a) **Mandatory requirements** constitute the core requirements which could be fulfilled with a GFM converter only and include self-synchronisation, phase jump active power, inertial active power, inherent reactive power, and positive damping power.

(b) **Optional requirements** constitute advanced GFM capabilities, e.g. black-start and those capabilities achievable by GFL

controls, e.g. sink for voltage unbalance and sink for harmonics.

(c) **Withstand capabilities** constitute the converters' abilities to withstand a large SCR change, phase jump, RoCoF, and temporary islanding.

## 3 Physical Limitations of OF-WPPs and Classification of GFM Requirements

### 3.1 Physical Capabilities and Limitations of GFM OF-WPPs

The GFM functional requirements reviewed in the previous section present multiple perspectives of the TSOs, expert groups, task forces, research institutes, and standards committees. The provided requirements are, in some cases, specific to generation types and, in other cases, general. In order to adapt the requirements into an OF-WPP application, the physical characteristics, capabilities and limitations of OF-WPP must be considered since the capabilities and limitations of OF-WPP (or any other generation source) impact different GFM functions, as is pointed out in EG-ACPPM (2023).

OF-WPPs with type-IV WTGs are connected to the grid via long AC transmission lines Johansen (2020) that have large impedances and low SCRs or via HVDC cables. The generation source itself is operated at its maximum power capacity or close to the maximum power capacity for the available resource at any instant via maximum power point tracking (MPPT) algorithms. Thus, OF-WPPs have limited headroom in their generation, which restricts the GFM operations such as phase-jump active power response and inertial response. The maximum overload capacity of the power-electronic switches of the converter

are significantly low compared to synchronous machines, for example around 1.11 pu of current injection Gomes Guerreiro et al. (2023). Even during the availability of wind and within the mechanical capacity of the WTGs, any overloading beyond





this can significantly stress and damage the power-electronic components of the WTG over its lifetime Erlich et al. (2009). Further, DC link energy availability Erlich et al. (2009) also pose a significant limitation on power transfer during transients, and thus also pose limitations on the GFM operation.

However, suppose the WTGs are utilised to curtail the available power for any given wind conditions. In that case, it provides some headroom for the GFM converter to react to any system events and provides the required GFM response, such as inertial response, phase jump active power response, or fault current contribution without overloading the power electronic switches Lyu and Groß (2024). This solution requires no hardware changes and thus can be achieved with simple software changes; however, it still imposes significant financial concerns on the generators as the curtailed power will not be transacted to the

grid. Furthermore, this also implies that the GFM behaviour of OF-WPP depends highly on the initial operating point and energy availability.

EG-ACPPM suggests using additional storage capacity at the DC link, fundamentally re-engineering the converter control, additional chopper capacity, over-dimensioning, and enhanced power headroom to enhance the GFM performance of the type-IV WTGs EG-ACPPM (2023). Although these solutions improve the technical capabilities in terms of the GFM requirements of

OF-WPPs, it must be noted that this implies significant hardware and control changes, thus putting significant financial concern on the manufacturers and developers, which can lead to a hike in the overall cost of the technology. Considering this, several TSOs (as discussed in the previous section) do not impose any hardware changes/additions to enhance GFM performances, which helps investigate more on the low-cost solutions to the GFM requirements. Therefore, any suggestions/conclusions on the need for significant hardware modifications/additions presented in this paper should be considered as a potential exploratory

path for further research and not as strict recommendations to implement them.

## 3.2 Classification of GFM Performance Specifications

Keeping the GFM requirements in Fig. 1 into due consideration and also taking into account the physical limitations of the WTGs and its associated mechanical and electrical components, a re-classification of the GFM performance specifications is performed in this section. An overview of the re-classification definition of mandatory, optional, and advanced GFM requirements is presented in Fig. 2.

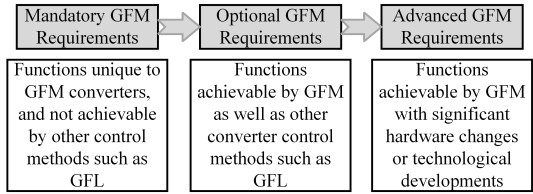

**Figure 2.** Re-classification of GFM functionalities: definition of the requirements/specifications categories.






### 3.2.1 Mandatory Performance Specifications

The GFM performance requirements, which are unique to GFM converters and can only be achieved by GFM converters, are included in this category. The performance specifications such as voltage source behaviour, synchronising active power, damping active power, inertial contribution, voltage support, FRT capabilities, control robustness, withstanding grid SCR changes, and RoCoF withstand are part of a majority of the technical documents reviewed in section 2. These requirements are the core capabilities expected of a GFM converter for any source, whether BESS, HVDC, or WPPs. OF WPPs especially have some restrictions and limitations on their inertial properties and overload capacity needed for requirements such as FRT capabilities, RoCoF withstand, and damping power. However, with certain power-boost features of the new OF WPPs, it is possible to explore these capabilities to some degree. Further, these requirements define the grid-forming behaviour at the most basic level and provide some necessary variations and updates from the grid-following behaviour. They also provide some similarities to some of the properties of the synchronous machines primarily sought after by IBRs. Thus, they are classified as mandatory requirements of gGFM OF WPPs. Further, some requirements are classified as one due to the nature of the responses and the use of the same test to assess them, namely, inertial response and RoCoF, and phase-jump/damping/synchronising active power.

### 3.2.2 Optional Performance Specifications

The optional performance specifications category involves such performance requirements, which could be an additional service to be provided by GFM converters, which other converter control methods such as V-f control or GFL control could provide. Typically, harmonics could be avoided by properly tuning filters or advanced techniques such as active filtering (Kocewiak et al., 2023). GFL converters could also cancel them and thus are usually classified as optional GFM requirements in the literature (InterOPERA, 2024). Although it is necessary for GFM converters not only to avoid injecting unnecessary harmonics but also to offer a sink for the harmonics arising from the grid, this requirement does not affect the core grid-forming behaviour and could be achieved with GFL converters; thus it is placed in the optional category of GFM requirement.

Furthermore, the requirement concerning the imbalance-withstand and providing a sink to grid imbalances by riding through unbalanced faults requires dedicated positive and negative sequence controls. These control realisations are not presented in this paper as they are not within the scope of the study, namely, the study of the dynamic behaviour of various OF WPP-based GFM converters against the performance specifications. This requirement is also not a core capability of GFM and can be realised in other primitive control methods such as GFL. Hence, it is classified as an optional requirement.

### 3.2.3 Advanced Performance Specifications

To provide extended inertia services, GFM converters need to be equipped with a reinforced DC bus, namely an energy buffer in the form of a flywheel energy storage system, a super-capacitor, or a battery (Rokrok et al., 2022). This technological extension increases the financial burden on OEMs, the inherent risks, and O&M requirements. Taking the consequences into



due consideration and acknowledging the need for extended inertia, this requirement is placed in the optional requirement category.

Islanding and auto re-synchronisation of OF GFM WPPs also present significant challenges, especially in the absence of a local load. The available energy following the islanding must be dissipated on the auxiliaries, the local loads, or via the DC chopper. In reality, the DC chopper has a limited capacity (Xu et al., 2021), the auxiliary consumption is typically low, and the local loads are absent. Thus, maintaining a steady open-circuit rated voltage during the islanding is challenging. Further, the auto re-synchronisation following the islanding can lead to a significant power surge as the phase angle between the PCC and the converter could differ. Achieving the capability of momentary islanded operation and auto-re-synchronisation following the

islanding potentially needs further studies on control strategies and hardware changes. Thus, this capability of GFM converters is categorised as an advanced requirement.

    Similarly, following the loss of the last synchronous generator in the grid, the GFM converter must autonomously run the entire grid by providing voltage and frequency set-points. This situation arises in a modern power system when most conventional synchronous generators are replaced by inverter-interfaced generation, and GFM IBRs run the power system. The

research field has some experience in running microgrids with GFM converters (Musca et al., 2022); however, the existing knowledge and experience are not adequate for a large power system as new challenges might arise with a high share of GFM IBRs in the system. As more studies, knowledge, and experience are required, assessing this requirement fulfilment is challenging. Considering the underlying challenges and lack of available studies on the subject matter, this requirement is categorised as an advanced requirement.

Black-starting a grid involves the gradual energisation of various power systems components, such as the transformer, transmission lines, and substations, thus building up the system voltage and a steady normal power flow. This area needs further research and field tests for OF WPPs as well as for other IBRs. Thus, achieving a black start with an OF WPP with GFM control is classified as an advanced performance.

## 4   Testing Grid-Forming Capabilities

### 4.1   Simulation-based Tests of GFM Specifications

The test framework of the GFM performances is a challenging topic to undertake as it requires a deep understanding of the GFM converters' performance requirements, capabilities, and significant practical experiences with specific generation sources. Most existing testing recommendations are for generators as a whole or specific generation type with no specific control method (GFL or GFM). Only a few TSOs have defined the testing requirements for GFM converters, which are also brief. Merely

following the existing testing requirements to build a test framework for OF GFM WPPs might not be relevant. Thus, this paper attempts to review the different test requirements provided by different grid codes and prepares relevant simulation-based test requirements for OF GFM WPPs. The goal of this section is not to numerically specify the test conditions but to provide a practical perspective on the extent of test severity which could be imposed on an OF GFM WPP simulation model.





**Table 1.** Recommended re-classification of GFM requirements for OF GFM WPPs and simulation-based tests to assess them.

| Classification | Requirement | Recommended Tests for GFM Functionality |
|---|---|---|
| Mandatory | Voltage source behaviour | Slowly changing voltage phasor during sub-transient (5-10 ms). The voltage should not vary sharply for any condition and must maintain a magnitude of 1 pu during the steady state. |
| | Synchronizing active power/damping active power | Synchronizing active power during grid phase jump, the initial operating point needs to be mentioned, and response to phase jump begins before 5 ms. |
| | Inertia contribution/RoCoF withstand | Active power response to grid RoCoF specified by the TSO. The inertial overshoot should not violate the converter's physical limits. |
| | Voltage support via reactive power | Reactive power response to grid voltage dip of 5 % from nominal. The converter should follow the V-Q droop property. |
| | Fast fault current contribution | Fault response should not rely on measurements. Response to faults should happen intrinsically and ideally begin before 5 ms from when the fault is applied. |
| | Withstand grid SCR changes | Operation at different SCR levels and SCR changes, maximum and minimum SCR levels must be defined at the PCC for each OF WPP. |
| Optional | Sink for harmonics | Reject grid voltage/frequency fluctuations |
| | Sink for imbalances | Ride through unbalanced faults; TSOs describe the fault severity levels. Converter operates within its hardware limitations. |
| | Interoperability | TD and FD studies for multi-GFM converter test system. Studies with black-boxed GFM WPPs with different GFM control methods are required to get a TSO perspective. |
| | Active power sharing/droop | Power delivery proportional to frequency change. |
| Advanced | Extended inertia via energy buffers | Assessed on case-by-case basis for generation types. |
| | Islanding and re-synchronization | Withstand islanding and auto re-synchronization for shorter islanding conditions. Local energy sink and re-synchronization schemes are needed to maintain stability for re-synchronisation attempts followed by longer islanding. |
| | Surviving the loss of last sync gen | Withstand sudden islanding with local loads. |
| | Black-start | Requires field test. |





Voltage source behaviour or a slow-moving internal voltage phasor is the core of GFM control and is required by many
GFM performance specifications. The voltage source behaviour could assessed by a passive impedance behaviour between the
5 Hz and 1kHz range (NG-ESO, 2019). Further, constant or nearly constant voltage internal voltage phasor in the sub-transient
time frame could be attributed to the good voltage-source behaviour of the GFM converter. Thus, the terminal voltage of
GFM converters should be constant or nearly constant in the transient and sub-transient time frame (<5-10 ms following the
disturbance). The disturbance could be a grid phase jump or a three-phase fault.

345    Energinet – the Danish TSO – has defined technical regulations for WPPs above 11 kW, which neither devote to specific
converter control methods of the WPPs nor to specific onshore or OF WPPs (Energinet, 2016). Nevertheless, the operational re-
quirements and tolerances for WPPs against different grid events at PCC are defined, and these could be adapted for simulation-
based tests on GFM converters. In contrast to the NG-ESO requirements of $30°$ phase jump tests and $\pm5\%$ grid voltage jump
tests for EMT simulation models (NG-ESO, 2024), Energinet specifies tolerance to $20°$ phase jump for 80-100 ms and $\pm20\%$
350    grid voltage jump for 0.5 s for WPPs with capacity higher than 25 MW. It must be noted that tolerance of smaller phase jumps
could be achieved by a GFM converter within its physical operation limits. However, for larger phase jumps, especially when
the GFM converter is operating near the maximum power, the phase jump can lead to instability, as shown in Fig. 3. Thus,
while defining phase-jump test requirements for GFM converters, the operating points, the available energy, and the energy
headroom must be considered.

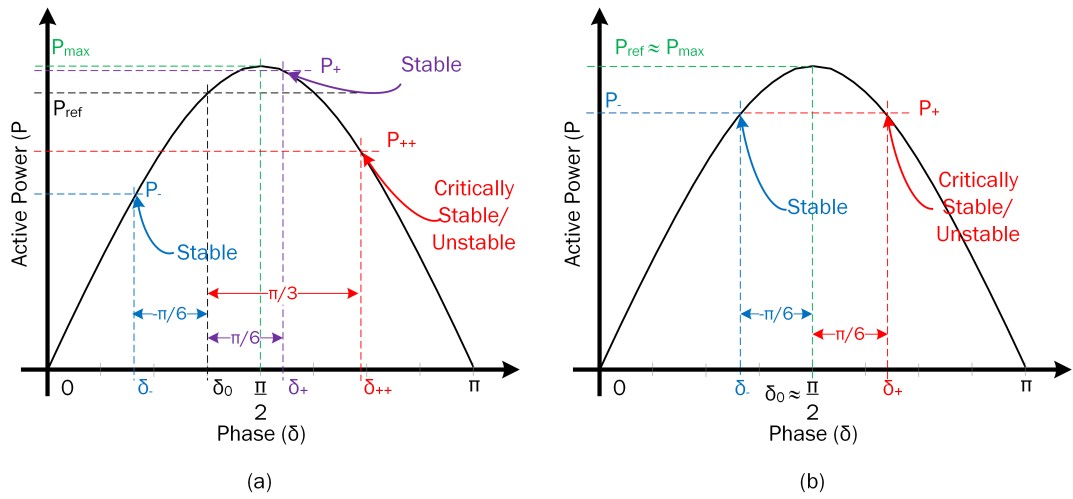

**Figure 3.** P-$\delta$ curves (illustration only) for large phase shifts: (a) for operating points below the rated power ($P_{ref} < P_{max}$), and (b) for
operating points close to the rated power ($P_{ref} \approx P_{max}$).

355    Energinet requirements also require WPPs to ride through a 150 ms phase-ground, phase-phase-ground, and three-phase
bolted faults for the WPPs (Energinet, 2016), which could be adapted for an OF GFM WPP; however, initial studies on the
FRT capabilities for GFM must be exploratory. However, a more crucial point to address for GFM regarding its fault response
is the inherent current capability it is required to possess. In response to any fault, the GFM should inherently start injecting



a fault current within its operational and hardware limits without relying on external measurements and control. The response start time needs to be within the sub-transient time frame and neither related to nor dependent on the measurement and control devices and their delays.

OF WPPs are characterised by weak grid connections with the SCR at PCC, typically below 2, which change dynamically during operation. Thus, a GFM OF WPP should be able to operate in a wide range of grid SCRs and withstand changes in these SCR values. Quantification of the specific values of SCR levels within which the OF GFM WPP is required to stay stable is not defined in this paper; however, it is recommended that an operating point sweep and stability analysis for a range of SCR values between its maximum and minimum is necessary to ensure that the GFM converter can provide stable operation within that range.

In the grid-code documents reviewed here, the requirements for islanding and re-synchronisation for GFM converters are not clearly stated, as is the case for other operational requirements. Topics such as islanding, re-synchronisation, black-start, and grid events such as the loss of the last synchronous generator in the grid are challenging to study and assess in simulation-based tests as they require extensive prototype and field testing. TenneT Annex C2.300 (TenneT, 2023) provides an islanding requirement of 150 ms for DC-connected PPMs, and InterOPERA plans to adapt this requirement and further explore the possibility of 300 ms islanding without enforcing any significant hardware changes on the PPMs (InterOPERA, 2024). However, both these requirements/proposals are for HVDC-connected PPMs following a DC fault and temporary blocking of the HVDC converters. They may not apply directly to AC-connected OF WPPs under their existing definition. Although OF GFM WPP could withstand shorter islanding situations in the order of a few tens of milliseconds, re-synchronization strategies might be required for longer islanding situations. Following a longer islanding situation, the converter phase and the grid phase can vary significantly. Without a proper re-synchronization strategy, the re-synchronization attempt can resemble a large grid phase shift, leading to instability. This is illustrated in figure 4. In some cases, due to the dynamic nature of the phase angles, the phase difference between the converter and the grid at the time of re-synchronization could also be aligned. However, this can still lead to issues such as over-burning of the DC chopper and large mechanical stress on the mechanical structure of the wind turbines ().

A summary is prepared for the proposed mandatory and optional GFM requirements for OF WPPs alongside the simulation-based tests to evaluate such requirements. It is presented in Table 1.

## 4.2 Emerging Test Benches for GFM tests

Traditionally, for WTGs and other equipment, grid compliance capabilities at the equipment level are performed using a full-scale prototype connected directly to the grid. A number of tests can be performed at this level, ranging from PQ capability curves, harmonics, flickering, under-voltage ride-through (UVRT), and over-voltage ride-through (OVRT) using a container test setup, etc. Nonetheless, this testing method can also offer many challenges in terms of testing limitations, safety, and limited freedom on changing grid conditions (e.g., frequency, SCR, and voltage).

Certain GFM requirements cannot be easily tested via traditional methodologies, such as equipment prototype turbines connected directly to the system or at the plant level (Gomes Guerreiro et al., 2024). Capabilities such as black-start, island op-

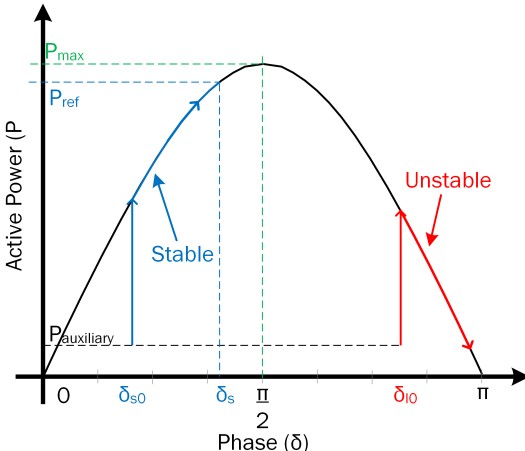

**Figure 4.** P-$\delta$ curves (illustration only) for short islanding (blue) and long islanding (red) cases followed by an auto-re-synchronization attempt.

eration, inertia response, RoCoF withstanding, etc., can be challenging to test for the first time on equipment/plants connected directly to the power system. Additionally, it is important to validate models for these additional capabilities. Thus, advanced

testing methods in a controllable environment, such as power hardware-in-the-loop (PHiL) with grid emulators either connected to subsystems or the prototype equipment, are emerging as important tools to assess GFM performances. Standardization of such test benches is also ongoing in technical fora, such as the IEC 61400-21-4 standard (IEC, Document 88/889/CD, 2023).

Various next-generation PHiL test benches have been proposed in (Curran et al., 2022; Neshati et al., 2023; Li et al., 2023; Gevorgian et al., 2023). These test benches use only a few selected components to represent the rest of the equipment by

employing novel modelling and emulation techniques for the missing hardware parts. These test benches primarily consist of the converter hardware and control system, which are connected to separate inverter systems capable of emulating either the generator and grid side or only the grid side. For example, (Neshati et al., 2023) presents a novel test rig with generator and grid-side emulators that can realistically emulate GFM capabilities. The generator-side emulator communicates with a real-time simulation of high-fidelity models of the WTG and aerodynamic components, which include wind field, turbulence,

and so on. The grid simulator uses an inverter-based control of the grid voltage to allow basic features such as grid FRT and advanced features such as dynamic impedance emulation, phase jumps, voltage steps, RoCoF, and harmonic control.

On the other hand, the use of grid emulators is also rising (Li et al., 2023; Hans et al., 2022, 2023) on tests performed at the equipment level, where instead of connecting the equipment directly to the system, grid emulators are connected to the equipment. This allows for a larger variety of tests and more controllability of testing conditions, as the grid emulator can

respond to any desired setpoints and operating conditions.



# 5  Conclusions

This paper reviews and summarises different grid codes, white papers, and technical documents on GFM functional specifications, operational requirements, and testing requirements. It also adapts the available functional specifications and operational requirements for OF GFM WPPs. It reclassifies them as GFM functionalities unique to GFM (mandatory requirements), GFM

functionalities achievable by other converter control methods (optional requirements), and advanced GFM functionalities that require hardware modification or significant technological advancements (advanced requirements). This paper also reviews various testing requirements for various generation sources and adapts them for OF GFM WPPs.

This paper also presents an overview of various other applications of GFM converters and recommendations on adapting the control requirements and functional specifications for these applications. Furthermore, a short review of the existing next-

generation test benches is provided with an outlook on their potential application for testing the capabilities of GFM converters for various generation sources.

It was observed from the rigorous literature review that the initial operating point for the GFM functional specifications is not defined. The behaviour of GFM OF-WPPs depends on their initial operating point and curtailment status; thus, the GFM standards need to specify these specifications on various initial operating points. Further, the standards need to consider the

limitations of OF-WPPs and the limitations of the switching devices. Finally, GFM specifications and requirements in grid codes must be technology-agnostic. Amidst the lacking literature, namely technical reports and performance specification recommendations of GFM converters for OF WPP application, this paper thus provides a framework to define and re-classify GFM requirements, recommend tests for evaluating such requirements, and also provide an overview of advanced test setups and how they could be utilized for the assessment of GFM behaviour.

*Author contributions.*  SG wrote the original draft. GGMG wrote section 4.2 and provided feedback for the rest of the sections. KVK and KHJ provided suggestions on recommended tests in Table 1 and also supported on the revision of the paper. KHJ and EDG provided guidelines and prepared the plans for the research as part of the funding application, and helped formulate paper contents and goal. GY, and XW contributed to the guidance and revisions.

*Competing interests.*  The authors declare no competing interests.

*Disclaimer.*  Figures and values presented in this paper should not be used to judge the performance of Siemens Gamesa (SGRE) technology as they are solely presented for demonstration purposes. Any opinions or analyses contained in this paper are the opinions of the authors and are not necessarily the same as those of SGRE.





*Acknowledgements.* The work of SG was supported by Innovation Fund Denmark under project Ref. no. 0153-00256B.



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
