# Peer review of "Functional Specifications and Testing Requirements of Grid-Forming Offshore Wind Power Plants"

_Wind Energy Science, 2024_

## Author Response (AR1)

**Original Manuscript ID:** wes-2024-61

**Original Article Title:** **"**Functional Specifications and Testing Requirements of Grid-Forming Offshore Wind Power Plants"

**To:** Editor,
Wind Energy Science Journal,
Special Issue on "Electro-mechanical interactions in wind turbines"

**Re:** Response to reviewers

Dear Editor,

Thank you for enabling such detailed and critical review of our manuscripts with multiple expert reviewers, and for providing us an opportunity to address the reviewers' comments.

We are uploading (a) our point-by-point response to the comments (below) (response to reviewers, under "Author's Response Files*"*), (b) an updated manuscript with yellow highlighting indicating changes (as "Highlighted PDF*"*), and (c) a clean updated manuscript without highlights ("Main Manuscript"*).*

Best regards,
Sulav Ghimire et al.

**RC#1:**

**Comment 1:** The paper is well-written and provide a  good overview of an increasingly important topic.

**Author response:**

Comment 1:  We thank the reviewer very much for the expert review and comments. We are glad that you found the work valuable and interesting.

**Author action: No action required.**
* * *
**CC#1/CC#2:**

**Comment 1: N/A.**

**Comment 2: It is a well-written paper with a good review of various test specifications suggested and/or mandated by different organizations and researchers. It is relevant to the current situation. Having a few simulation examples to support the effectiveness of the suggested tests for GFM could strengthen the effectiveness of the paper.**

**Author response:**

Comment 1: The authors would like to thank the reviewer for his kind input. All of the inputs provided in the pdf file regarding typographical and grammatical errors were addressed and will be updated with the new version of the manuscript.

Comment 2: Thank you very much for your expert comments. They are greatly appreciated. The idea to integrate some results and comparison of various GFM control technologies into this manuscript was thought of at the beginning of the writing phase; however, after rigorous brainstorming, we, the authors, decided to dedicate this manuscript to the review and re-classification of GFM functional specifications for offshore WPP use and to work on another manuscript for the simulation results and comparison. This decision allowed us, the authors, to focus on the specific goals of each of these manuscripts and prevented our work from being too cluttered. Thus, we kindly request the reviewer to wait for our upcoming work which will compare various representative GFM control methods based on the performance specifications and testing methods suggested in this manuscript. In the meantime, some preliminary work done by us on this subject is available at: Grid-forming control methods for weakly connected offshore WPPs | IET Conference Publication | IEEE Xplore.

**Author action: [The typographical errors have been corrected.]**
* * *
**CC#2:**

Comment 1: The research topic is very interesting and obviously there is a distinguished effort made by the authors to review the state of the art of the grid codes and guidelines of the Grid Forming requirements and testing methodologies. The paper evaluates and summarize perfectly the new tendencies and most relevant capabilities of the new control methodology.

I recommend checking the spelling of the document to leave it correctly because there are numerous spelling errors along the document.

**Author response:**

Comment 1: The authors would like to thank the reviewers for his/her comments. In the edited version of the manuscript, the authors have carefully checked the spelling and other grammatical errors and made necessary adjustments.

**Author action: [The typographical errors have been corrected.]**
* * *
**RC#3:**

Comment 1: The research topic of the paper is really interesting. The paper is well written and the statements are clearly exposed and explained. The authors made a very nice effort to bring all these existing contributions, especially regarding the grid forming and testing methodologies.

I think some minor corrections could be made by a careful reading of the paper, some typing mistakes still exist and some sentences could be improved in order to provide the clearest point of view.

**Author response:**

Comment 1: The authors would like to thank the reviewers for his/her comments. In the edited version of the manuscript, the authors have carefully checked the spelling and other grammatical errors and made necessary adjustments. Further, the authors have updated the manuscript by re-formulating the long sentences and clarifying the statements when necessary.

**Author action: [The typographical errors and sentence clarity issues have been corrected.]**
* * *
**RC#4:** The paper provides an excellent overview of the existing requirements/grid codes/testing specifications for grid forming capability. The paper is clear and well written. I only have a few relatviely minor comments.

Comment 1: I'd suggest the titles of GFM requirements on Figure 1 are clarified in the text. For example it is unclear how the distinction is made between voltage jump reactive power respose and fast fault current contribution. Not all requirements have this separation. For phase jump and RoCoF requirements it is unclear if "withstand" requirement is included under same category or not? It's not clear what's ment by "Active power sharing/power dispatch", I would expect this requirement to be there explicitly or implicitly in every spec?

Comment 2: Some specifications explicitly call out some requirements once a resource reaches it's limits, but there is no box for this in Figure 1.

Comment 3: Also in Figure 1, AEMO is missing in "withstand SCR changes" box. There is not such specific requirements in their specs but there is another document from AEMO on grid forming testing that came out about 8 month later, that has this test for GFM and the expectation is to withstand.

Comment 4: In Figure 1 for Surviving the loss of last synchronous machine  NERC requirement is missing. There is a test for this in NERC white paper

Comment 5 Lastly I don't understand why IEEE 2800 is listed. It is not specific to GFM and has no additional requirements for GFM. The standard was developed with GFL IBRs in mind, even though it doesn't call out GFL specifically. I would suggest just removing it from Figure 1 and from main text to avoid confusion.

Comment 6: Are the limitations that atre listed in Section 3, only for offshore wind or wind technology overall. I suggest that clear distinction is made between the limitations that apply to all wind generation and those that only apply for offshore wind plants.

**Author response and Author actions:** The authors thank the reviewer for his/her detailed comments and suggestions. They were very relevant and helpful. The authors have attempted to make changes to the manuscript accordingly and provided answers to the reviewer's comments. The details of these changes are marked in the marked manuscript file, and are also summarized herewith.

Comment 1: Explanations for these comments are provided in the paper. A summary of the changes in the paper is provided below.

The titles from Figure 1 are explained in the text. Brief introductions to the titles of GFM requirements are included in the paper with an explanation that these introductions are generic and are based on the documents reviewed; however, not specifically limited to them.

Withstand requirement is placed under the same category. The only place where it varies is the operating modes, namely normal operating mode, withstand operating mode, and emergency operating mode. Under normal operating mode, GFM OF WPPs should ride-through the events such as RoCoF or phase jumps. For withstand mode, GFM OF WPP should try to withstand such events, not necessarily ride-through them. On the other hand, the emergency operating mode is critical, and GFM OF WPP's response to events during this mode could range from ride-through, withstand, or trip based on the grid-code suggestions.

Yes, the topic of active power sharing/power dispatch is implicit or explicit in all the documents; however, it has been included in the paper as it is important to distinguish cases where power-sharing by GFM devices is of great interest, especially cases where IBRs dominate the power system. Another reason behind mentioning this topic explicitly in the paper is to clarify the interoperability and power-sharing with other WPPs connected in electrical proximity in a multi-vendor power system.

Comment 2: We agree with the reviewer's comments. Some specifications mention current-limited operation and some requirements during such a condition. However, our work related to the normal operating condition and current-limited or withstand operation is out of the scope of the work. Thus, they are not included in the paper.

Comment 3: The suggested point has been added. The document mentioned by the reviewer was not yet released when this manuscript was first written; however, its short review and summary in Figure 1 have now been included in the paper.

Comment 4: The suggested change has been added to the figure.

Comment 5: The explanation to this has been provided in section 2.9. We have tried to further clarify it in the same section. "Although the standard generally discusses IBRs, the insights on the interoperability of IBRs and the IBR testing requirements it provides are crucial for adaptation to GFM OF WPPs. As is the case for the other documents reviewed -- where GFM specifications are defined either as agnostic of the generation sources or for specific generation sources that are not OF WPPs -- we have reviewed and adapted the specifications and testing suggestions for GFM OF WPP uses. Similarly, we adapt some generic IBR requirements and testing suggestions for GFM OF WPPs in this paper."

Comment 6: The limitations discussed in the paper are only limited to offshore wind technology, however, some limitations can also apply for onshore wind. This clarification has been made in the paper.